# Comparative Evaluation of Intradermal *vis*-*à*-*vis* Intramuscular Pre-Exposure Prophylactic Vaccination against Rabies in Cattle

**DOI:** 10.3390/vaccines11050885

**Published:** 2023-04-23

**Authors:** Swathi Gopalaiah, Kshama M. Appaiah, Shrikrishna Isloor, Dilip Lakshman, Ramesh P. Thimmaiah, Suguna Rao, Mahadevappa Gouri, Naveen Kumar, Kavitha Govindaiah, Avinash Bhat, Simmi Tiwari

**Affiliations:** 1Department of Veterinary Medicine, Veterinary College, KVAFSU, Bengaluru 560024, India; 2KVAFSU-CVA Rabies Diagnostic Laboratory, WOAH Reference Laboratory for Rabies, Department of Veterinary Microbiology, Veterinary College, KVAFSU, Bengaluru 560024, India; 3Department of Veterinary Pathology, Veterinary College, KVAFSU, Bengaluru 560024, India; 4Department of LFC, Veterinary College, KVAFSU, Bengaluru 560024, India; 5Department of AGB, Veterinary College, KVAFSU, Bengaluru 560024, India; 6Department of Biological Production, IAH and VB, KVAFSU, Bengaluru 560024, India; 7Masterlab, Nutreco, 5831 JN Boxmeer, The Netherlands; 8Zoonosis Division, National Centre for Disease Control, Directorate General of Health Service, Ministry of Health and Family Welfare, GOI, Delhi 110054, India

**Keywords:** pre-exposure prophylactic vaccine, cattle, ID, IM, RFFIT, RVNA

## Abstract

Rabies is a progressively fatal viral disease affecting a wide variety of warm-blooded animals and human beings. With cattle being major part of Indian livestock population, rabies can result in significant financial losses. Immunization of livestock vulnerable to exposure is the best way to control rabies. The present study was undertaken to investigate the efficacy of a rabies pre-exposure prophylactic vaccine administered through different routes and to sequentially monitor the levels of rabies virus-neutralizing antibody (RVNA) titers in cattle. Thirty cattle were divided into five groups of six animals each. Group I and III animals were immunized with 1 mL and 0.2 mL of rabies vaccine through intramuscular (IM) and intradermal (ID) routes, respectively, on day 0, with a booster dose on day 21; Group II and IV animals were immunized with 1 mL and 0.2 mL of rabies vaccine, respectively, without the booster dose; unvaccinated animals served as a control (Group V). Serum samples were collected on days 0, 14, 28, and 90 to estimate RVNA titers using the rapid fluorescent focus inhibition test (RFFIT). The titers were above an adequate level (≥0.5 IU/mL) on day 14 and maintained up to 90 days in all animals administered the rabies vaccine through the IM and ID route with or without a booster dose. The study indicated that both routes of vaccination are safe and effective in providing protection against rabies. Hence, both routes can be considered for pre-exposure prophylaxis. However, the ID route proved to be more economical due to its dose-sparing effect.

## 1. Introduction

Rabies is one of the most important zoonotic diseases caused by a lyssa virus of the family *Rhabdoviridae*, which results in a progressively fatal and incurable viral encephalitis. It affects human beings, as well as several species of animals. Rabies is endemic to India, and approximately 25,000–30,000 human deaths due to rabies are being reported annually as per the WHO [1,2]; it is also estimated that 35% of the reported global deaths due to rabies occur in India [3]. The majority of people who die of rabies are those belonging to the economically weaker section of the society. Despite the high incidence of rabies in India, it was until very recently not a notifiable disease; hence, a structured surveillance system is not in place [4,5]. There is no organized surveillance system for human cases (much less animal cases) of rabies, and the actual number of deaths due to rabies could be much higher than reported [1,6,7,8,9]. Dog bites account for 97% of human rabies and are probably a major cause of rabies in livestock [2,10]. Livestock, especially cattle, play a significant role in the life and economy of the people in India, where 70% of the total population is dependent on agriculture. India has the highest cattle population in the world; as per the 2019 livestock census, the cattle population in India was 145.91 million. Because cattle are such a major part of the Indian livestock population, viral diseases such as rabies can cause significant economic losses to the farmers through deaths and loss of production. However, in India, there are no reliable reports on rabies-related cattle deaths, and it is mostly under reported as with rabies-related human deaths; hence, the exact data on deaths due to rabies in cattle are not available. It has been reported that deaths due to rabies are grossly underestimated [6]. Furthermore, there is also the risk of human exposure as most of the cattle owners in rural India belong to the economically poorer section without access to proper education. A study in Bhutan on rabies awareness among cattle owners found that there was a lack of comprehensive knowledge about rabies, such as susceptible hosts, transmission route, clinical signs, and prophylactic measures, thus posing a potential public health threat to humans during handling of rabid animals and the carcasses [11]. Moreover, in rural India, cattle are allowed to graze free, and there is a constant interaction among livestock, humans, and dogs (pets and strays); thus, cattle are always at risk of exposure, and the owners on several instances may be unaware of it. This makes exposure to rabies a very real possibility not only for cattle but also for humans.

Nevertheless, rabies is an entirely preventable disease, and it can be prevented by post-exposure prophylactic vaccination [10,12]. However, vaccination of humans for prevention of rabies comes at a very substantial cost, and developing countries such as India will not be able to afford it. Hence, rabies control through mass vaccination of stray dogs has consistently proven to be cost-effective in preventing deaths due to rabies in humans [3,12]. Toward this end, the World Health Organization (WHO), World Organization of Animal Health (OIE), Food and Agriculture Organization (FAO), and Global Alliance for Rabies Control (GARC) have set a global goal of elimination of dog-mediated human rabies by 2030. In view of this, India launched the “National Action Plan for Rabies Elimination” in 2021 and is working toward achieving complete elimination of rabies. Hence, prophylactic vaccination of livestock would be a step forward in the right direction in areas with a high population of stray dogs, and a simultaneous effort in the elimination of rabies in cattle and other livestock can contribute significantly toward achieving this goal. To further reduce the vaccination cost and simultaneously address the issue of vaccine shortage, the WHO reviewed the cell culture rabies vaccine (with a potency > 2.5 IU per intramuscular dose) administered via the ID route at a much lower dose for post-exposure prophylaxis (PEP) or pre-exposure prophylaxis (PrEP); they found that the efficacy was comparable to or greater than the same vaccine administered via the IM route in humans [2,10]. Similar studies have been carried out in dogs, showing promise. However, there is paucity of information on the efficacy and feasibility of employing the rabies pre-exposure prophylactic vaccination through the intradermal route in cattle, which, if found efficacious, could reduce the cost of the vaccine, thereby achieving the goal of complete prophylactic vaccination in cattle as a part of the initiative to eliminate rabies. In view of the above, the present study was undertaken to investigate the most efficacious method of rabies vaccination and to estimate the level of rabies virus-neutralizing antibody (RVNA) in cattle through different routes (ID and IM) and doses using Nobivac^®^ Rabies vaccine.

## 2. Materials and Methods

### 2.1. Study Animals

The animals selected for the study were healthy cattle that were not pregnant and were between second and sixth lactation; they were housed at the Livestock Farm Complex, Veterinary College, Bengaluru, which had not been previously vaccinated against rabies. The study protocol was designed as per the standards of the Institutional Animal Ethics Committee (IAEC). The study protocol was proposed to the IAEC, Veterinary College, Hebbal, Bengaluru and received IAEC approval (approval number VCH/IAEC/2021/32, dated 27 July 2021).

### 2.2. Vaccine

Nobivac^®^ Rabies vaccine containing an inactivated Pasteur RIVM rabies strain with potency ≥ 2 IU per mL was used in this study. The vaccine was manufactured by MSD Animal Health, and the adjuvant used was 2% aluminum phosphate.

### 2.3. Study Protocol and Sample Collection

The selected cattle were divided into five groups of six animals each. Animals in Group I were immunized with 1 mL (as recommended by manufacturer) of rabies vaccine (Nobivac^®^ Rabies, MSD Animal Health, NJ, USA) intramuscularly in the neck region on day 0 and a booster vaccine on day 21. Animals of Group II were immunized with 1 mL of rabies vaccine intramuscularly in the neck region on day 0 without a subsequent booster dose. Animals of Group III were immunized with 0.2 mL of rabies vaccine intradermally at the middle third of the neck region on day 0 and a booster vaccine on day 21 as per the studies of [13,14,15]. Animals of Group IV were immunized with 0.2 mL of rabies vaccine intradermally at the middle third of the neck region on day 0 without a subsequent booster dose. Intradermal (ID) administration of rabies vaccine was performed using a tuberculin syringe (1 mL), and successful vaccination was confirmed by the appearance of a raised papule or bleb about 1–1.5 cm, which appeared immediately after vaccination (Figure 1). Cattle in group V were not vaccinated and maintained as a control. Two milliliters of blood was collected under sterile precautions on day 0 prior to the vaccination and again on day 14, 28, and 90 post vaccination from all vaccinated animals. The serum was separated by centrifuging the blood at 5000 rpm for 5 min and was stored at −20 °C; the RVNA titer was estimated by RFFIT.

### 2.4. Rapid Fluorescent Focus Inhibition Test (RFFIT)

In this study, the RFFIT protocol standardized at the KVAFSU-CVA Rabies Diagnostic Laboratory, WOAH Reference Laboratory for Rabies, Department of Veterinary Microbiology, Veterinary College, Bengaluru, was used for all serum samples [16,17]. The serum samples were incubated in a water bath for complement inactivation at 56 °C for 30 min. Later, the samples were serially diluted in a flat-bottomed 96-well microtiter plate (Nunc MaxiSorp™ flat-bottom, Thermo Fisher Scientific, Waltham, MA, USA). Following this, 100 TCID_50_ of rabies virus was added to all wells except cell controls, and then incubated at 37 °C for 90 min. Furthermore, approximately 25,000–30,000 BHK-21 cells/well were seeded into all wells of the plate and incubated at 37 °C for 48 h in a 5% CO_2_ incubator. After the incubation, the medium was decanted from the plate without disturbing the monolayer, and the cells were fixed with 70% chilled acetone for 30 min at −20 °C. The fixed cells were incubated with a 1:5 diluted fluorescein-labeled anti-RABV nucleoprotein antibody (Fujirebio Diagnostics, Malvern, PA, USA) for 1 h at 37 °C. The plates were examined using a fluorescent microscope and observed for fluorescent foci. The titer of RVNA was estimated in comparison with the WHO reference serum.

### 2.5. Estimation of RVNA Titers by RFFIT

Serum samples collected were tested for neutralizing antibody titers against rabies by RFFIT, and the plates were observed under an inverted fluorescent microscope (with excitation filters of 480 nm and emission at 530 nm). The highest dilution of the serum which caused complete neutralization of RABV was considered for estimation of the RVNA titer. In this study, an RVNA titer of 0.5 IU/mL serum was considered to possess an adequate immune response. The titer was estimated using the following equation:$$\mathrm{Rabies}\;\mathrm{virus}\;\mathrm{neutralizing}\;\mathrm{antibody}\;\mathrm{titer}\;\mathrm{in}\;\mathrm{test}\;\mathrm{serum}\;\left(\mathrm{IU}/\mathrm{mL}\right)=\frac{\mathrm{Reciprocal}\;\mathrm{of}\;\mathrm{highest}\;\mathrm{dilution}\;\mathrm{test}\;\mathrm{serum}\;\mathrm{showing}\;\mathrm{complete}\;\mathrm{neutralization}\;\mathrm{of}\;\mathrm{virus}\;\mathrm{infectivity}\;\times \;\mathrm{unit}\;\mathrm{of}\;\mathrm{reference}\;\mathrm{serum}}{\mathrm{Reciprocal}\;\mathrm{of}\;\mathrm{highest}\;\mathrm{dilution}\;\mathrm{of}\;\mathrm{reference}\;\mathrm{serum}\;\mathrm{showing}\;\mathrm{complete}\;\mathrm{neutralization}\;\mathrm{of}\;\mathrm{virus}\;\mathrm{infectivity}}$$

### 2.6. Statistical Analysis

The data obtained in this study were subjected to statistical analysis using two-way ANOVA with the help of a computer-based statistical program using R Studio version 1.4.1103 with Base R version 4.0.4, packages deplyr, tidyverse, tidyr, plotrix, and ggplot2 to arrive at the appropriate conclusion.

## 3. Results

### 3.1. Sequential Monitoring of RVNA Titer to Detect the Efficacy of Rabies Pre-Exposure Prophylactic Vaccine Administered through Different Routes

The serum samples without RVNA resulted in growth of virus, wherein the fluorescent foci were visualized under an inverted fluorescent microscope (Figure 2). However, the serum samples containing RVNA did not show any fluorescent foci, as the virus was neutralized by the antibodies in the test/reference serum (Figure 3).

All six cattle of all four groups (I, II, III, and IV) showed RVNA titers of above 0.5 IU/mL (Table 1). The geometric mean (GM) ± geometric standard deviation (GSD) (IU/mL) of RVNA titer in Group I was found to be 0, 2.00 ± 2.40, 22.63 ± 5.08, and 8.98 ± 5.91 on days 0, 14, 28, and 90, respectively (Table 2 and Figure 4A). The GM ± GSD (IU/mL) of RVNA titers in Group II was found to be 0, 11.31 ± 3.39, 40.32 ± 1.76, and 16.00 ± 2.67 on days 0, 14, 28, and 90, respectively (Table 2 and Figure 4B). The GM ± GSD (IU/mL) of RVNA titers in Group III was found to be 0, 2.52 ± 1.76, 40.32 ± 1.76, and 12.70 ± 3.10 on days 0, 14, 28, and 90, respectively (Table 2 and Figure 4C).The GM ± GSD (IU/mL) of RVNA titers in Group IV was found to be 0, 3.18 ± 2.05, 5.04 ± 2.32, and 2.00 ± 2.40 on days 0, 14, 28, and 90, respectively (Table 2 and Figure 4D). The seroconversion was found to be 100% in all four groups (I, II, III, and IV) on days 14, 28, and 90. 

### 3.2. Comparison of Efficacy of Rabies Vaccine Administered via Different Routes (Group I, II, III, and IV) Based on RVNA Titers

The efficacy of rabies vaccine administered via different routes and the level of RVNA titer were compared in groups I, II, III, IV, and V (control) on days 14, 28 and 90.

On day 0, RVNA titers in serum collected on day 0 were found to be below detectable levels (<0.5 IU/mL) in all four groups and the control (Group V).

On day 14, RVNA titers ranged from 0.5 to 4 IU/mL, 2 to 64 IU/mL, 1 to 4 IU/mL, and 1 to 8 IU/mL in Group I, Group II, Group III, and Group IV, respectively (Table 1, Figure 5). All cattle (100%) in all four groups (I, II, III, and IV) developed antibody titers above an adequate level (≥0.5 IU/mL). Among the four groups (I, II, III, and IV), no significant difference (*p* > 0.05) in RVNA titers was observed (Table 2).

On day 28, RVNA titers ranged from 1 to 64 IU/mL, 16 to 64 IU/mL, 16 to 64 IU/mL, and 1 to 8 IU/mL in Group I, Group II, Group III, and Group IV, respectively (Table 1). All te cattle (100%) in all four groups (I, II, III, and IV) developed antibody titers above an adequate level (≥0.5 IU/mL). Among the four groups (I, II, III, and IV), a significant difference (*p* < 0.05) in RVNA titers was noticed between Group I and Group IV, between Group II and Group IV, and between Group III and Group IV (Table 2). The highest mean RVNA titer for the intramuscular and intradermal route was seen on day 28.

On day 90, RVNA titers ranged from 1 to 64 IU/mL, 4 to 64 IU/mL, 4 to 64 IU/mL, and 1 to 8 IU/mL in Group I, Group II, Group III, and Group IV, respectively (Table 1). Among the four groups (I, II, III, and IV), no significant difference (*p* > 0.05) in RVNA titers was noticed (Table 2).

All four groups of animals vaccinated against rabies via different routes (IM and ID) with and without booster had developed adequate levels of RVNA well over the protective levels required (≥0.5 IU/mL) by day 14, and it was maintained until the end of the study period, i.e., 90 days, irrespective of route of vaccination. The geometric mean with geometric standard deviation of RVNA titers of groups I, II, III, IV, and V is depicted in Figure 5, and the range of RVNA titers in different vaccination groups tested sequentially over a period of time from day 0 to day 90 is depicted in Figure 6.

## 4. Discussion

Rabies is a fatal encephalitis caused by rabies virus (RABV) and is a disease affecting humans and animals, both domestic and wild. Although vaccines are available to contain the disease, there are still several reports every year of human deaths due to rabies. Hence, efforts are required on a war footing to contain this deadly, lethal disease, and prevent its further spread. It is reported that, across the globe, particularly in Africa, Asia, and India, domestic dog-adapted RABV wages a neglected epidemic that claims an estimated 59,000 human lives annually [18].

Human deaths due to rabies are mostly attributed to bites from dogs, foxes, raccoons, skunks, jackals, mongooses, and other wild carnivores, although rare. However, domestic livestock such as cattle and buffaloes is not the major source of transmission of rabies to other hosts including human beings but such livestock can be the victims of rabies. This could lead to the crippling of economy of the rural livestock owner/farmer or dairy industry. Hence, prophylactic vaccination of cattle can be considered provided it is viable and cost-effective. Every year, more than 29 million people worldwide receive rabies post-exposure prophylaxis (PEP), and the average cost is estimated to be 8.6 billion USD per year [6]. Eradication of the disease entirely depends on successful pre- and post-exposure vaccination. Toward this end, the WHO, in its roadmap for neglected tropical diseases, included rabies as one of the targeted diseases. India, in its 12th five-year plan, envisaged rabies control as a priority area and is working toward achieving this goal, launching the “National Action Plan for Dog Mediated Rabies Elimination from India by 2030 (NAPRE)”. Canine vaccination is the suggested strategy of choice toward achieving this objective [9]. Intradermal administration of rabies vaccine for post-exposure prophylaxis in humans has resulted in a major paradigm shift in rabies control programs in humans. WHO recommends this route (i.e., ID) when vaccines are in short supply, citing a 60–80% reduction in direct costs and vaccine consumption when compared with the intramuscular route [9]. A study on intradermal rabies vaccination emphasized the need to adopt the most immunogenic, economical, and acceptable method of vaccination to achieve greatest benefit for communities in dog rabies-endemic countries [19].

However, in dogs and other animals, this is a vastly unexplored area; of late, there are a few research publications endorsing the efficacy of the intradermal route of pre-exposure and post-exposure prophylaxis for the control of rabies in dogs. Furthermore, adapting this route of vaccination in dogs would be highly cost-effective as the same efficacy can be achieved with one-fifth of the intramuscular dose, thus representing a more economically viable and attractive option. In India, being a rural economy, the farmers are largely dependent on agriculture and livestock such as cattle, buffaloes, sheep, goats, and pigs for livelihood, and they are intimately associated with these animals, along with dogs, which are generally kept as pets or guard dogs. Hence, there is a constant interaction among humans, livestock and dogs. Most of the cattle and other livestock in India, unlike in developed countries (stall-fed), are free-ranging and allowed to graze on available pasture and forest areas; thus, they are always at risk of rabid dog bites. Furthermore, most of these bites can go undetected and can pose a serious threat to animal and human lives. Hence, vaccinating these animals should also be an essential part of the strategy to control rabies. Several studies in dogs have found the intradermal route of vaccination to be as effective as the intramuscular route with the added benefit of cost-effectiveness, as only one-fifth of the IM/SC dose is administered intradermally [20,21].

This study was conducted in cattle to detect and compare the antibody titers following anti-rabies pre-exposure prophylactic vaccination via intramuscular and intradermal routes with and without a booster dose using the Nobivac^®^ Rabies vaccine. The dose used for the intradermal route was one-fifth of the intramuscular route (0.2 mL). Several workers used this dose in dogs and achieved adequate antibody titers [20,21,22]. It is an established fact that the dermis and epidermis are richly endowed with various types of dendritic cells that are capable of stimulating both an innate and an adaptive immune response [23,24,25]. This makes it an ideal target site for vaccine administration.

The rapid fluorescent focus inhibition test was employed to measure the serum neutralizing antibody titers. It is the standard assay recommended by the WHO for determining the levels of RVNA, and it was found to have good specificity and repeatability [26]. In this study, an effort was made to check if an adequate immune response could be achieved in cattle similar to that reported for dogs via the intradermal route of anti-rabies PrEP vaccination using Nobivac^®^ Rabies vaccine, which is the most popular vaccine used in dogs in India because of its good immune response. There are other studies on intradermal PrEP vaccination in cattle but none of the studies used Nobivac^®^ Rabies as the vaccine of choice. This is especially relevant in view of the fact that MSD Intervet (manufacturers of Nobivac^®^ Rabies vaccine) has been identified as one of the new suppliers for the WOAH rabies vaccine bank [27]. Hence, this study also aimed to check if there was better immune response as indicated by antibody titers for this vaccine as compared to other vaccines such as Raksharab and Rabivac Vet, used by other workers [14,15,28].

In the present study, it was found that all four groups (namely, intramuscular with and without booster (Group I and II) and intradermal with and without booster (Group III and IV)) were able to achieve adequate antibody titers above the recommended level of 0.5 IU/mL by day 14 following the primary vaccination, which reached a peak on day 28, followed by a gradual decline, but adequate levels were maintained until the end of the study period (i.e., day 90). The antibody response was comparable to the intramuscular route in the present study. A study using Rabivac Vet observed adequate levels of immunity following a single ID (day 0) injection of 0.2 mL of rabies vaccine, which was comparable to two doses (days 0 and 3), three doses (days 0, 3, and 7), four doses (day 0, 3, 7, and 14), and five doses (day 0, 3, 7, 14, and 28) of rabies pre-exposure prophylactic vaccine [15]. Another similar study using Raksharab vaccine found an adequate immune response by day 14 in both IM and ID routes with a marginally better response in animals vaccinated via the IM route; the response via ID route peaked on day 28 and reached levels comparable with IM route [14]. However, a study in Bhutan using Raksharab as the vaccine of choice with the same protocols as the present study observed a better response via the intramuscular route as compared to the intradermal route, with an adequate immune response in 71% of cattle vaccinated intradermally vs. 89% in cattle vaccinated via the intramuscular route; the response was only 36% and 58% in ID vaccinated animals on days 14 and 30, respectively [28]. Thus, there are varied reports about the immune response to Raksharab vaccine used intradermally. Hence, the present study is of relevance as there are no reports of Nobivac^®^ Rabies being used in any of the earlier studies in cattle. Furthermore, it was observed that a booster dose, especially in the animals vaccinated via the intradermal route, increased the level of protection conferred on the animal and probably ensured a more prolonged immune response. Administration of a booster for pre-exposure prophylactic vaccination for rabies in cattle resulted in high antibody titers as expected [29,30,31]. In the present study, adequate immune titers (>0.5 IU/mL) were maintained in all the four groups until the completion of the study (i.e., 90 days). An earlier study reported that there were adequate antibody titers for 180 days following the intradermal route of rabies vaccine administration without a booster [13]. Furthermore, most of the previous studies [14,15,28] on intradermal rabies used rabies vaccines containing aluminum hydroxide as an adjuvant (Raksharab, Rabivac Vet), whereas the vaccine used in the present study had aluminum phosphate as the adjuvant. Whether this has any effect on the immune response needs to be ascertained, and the vaccine containing the most immunogenic adjuvant may be advocated for single intradermal prophylactic vaccination. An analysis of the immune cells at the site of injection and proteome analysis of the muscle tissue revealed differentially regulated processes related to innate immune response between the two [32]. In vivo studies showed that immunization with aluminum hydroxide as the adjuvant attracted neutrophils, whereas aluminum phosphate attracted monocytes/macrophages to the site of injection; both adjuvants acted differently on the innate immune system [32]. However, whether one is superior to the other in terms of the immune response induced needs to be ascertained, and further research may be warranted to arrive at the most useful adjuvant to be incorporated in the rabies vaccine for better efficacy.

## 5. Conclusions

The intradermal route for administration of anti-rabies pre-exposure prophylaxis vaccine to cattle at 0.2 mL using Nobivac^®^ Rabies vaccine was found to be effective in achieving adequate immunity. Hence, a single intradermal pre-exposure prophylactic vaccination can be a viable, cost-effective option for vaccinating not only dogs but also cattle, especially in rabies-endemic countries of Asia and Africa. However, several other factors such as the adjuvant used and the site of administration of the intradermal vaccine have an important role to play in determining the extent of response achieved, and further studies may be warranted to achieve the most practical, efficacious, and cost-effective option.

## Figures and Tables

**Figure 1 vaccines-11-00885-f001:**
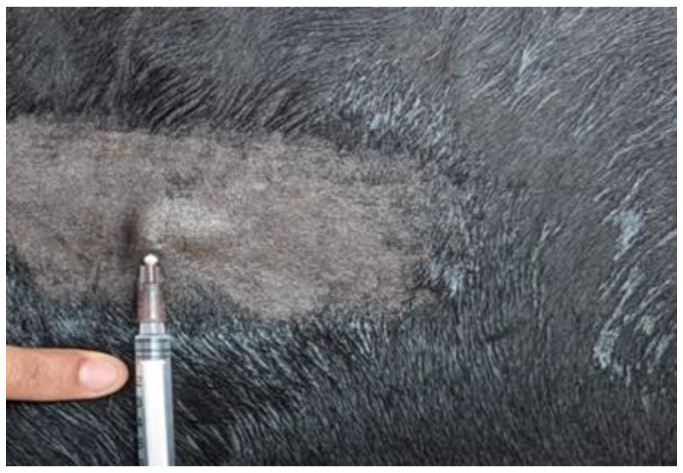
Intradermal route of vaccination with bleb formation.

**Figure 2 vaccines-11-00885-f002:**
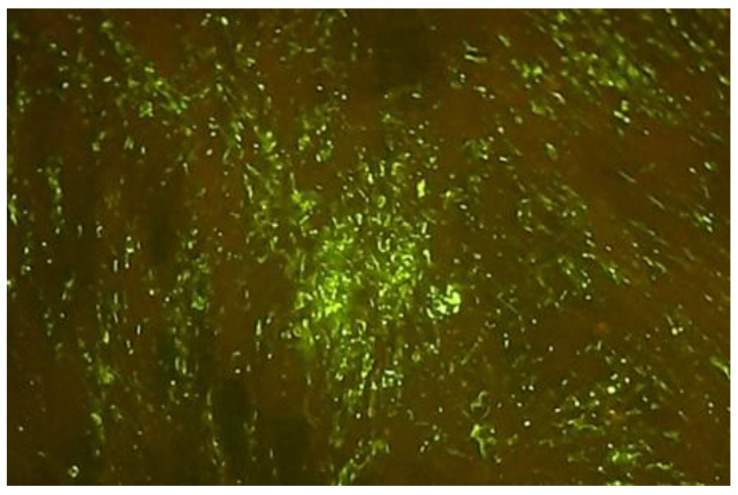
Fluorescent foci from a well with virus control of microtiter plate (100 TCID_50_).

**Figure 3 vaccines-11-00885-f003:**
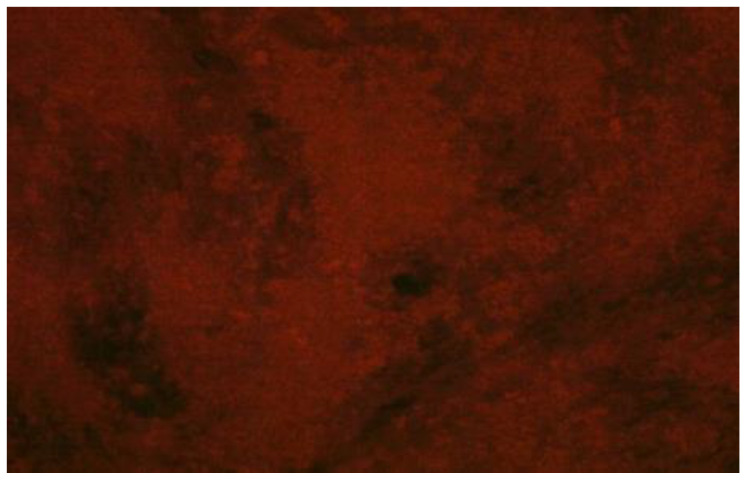
Absence of fluorescent foci from a microtiter plate well showing complete neutralization of virus in post-vaccinal serum from an animal of group III.

**Figure 4 vaccines-11-00885-f004:**
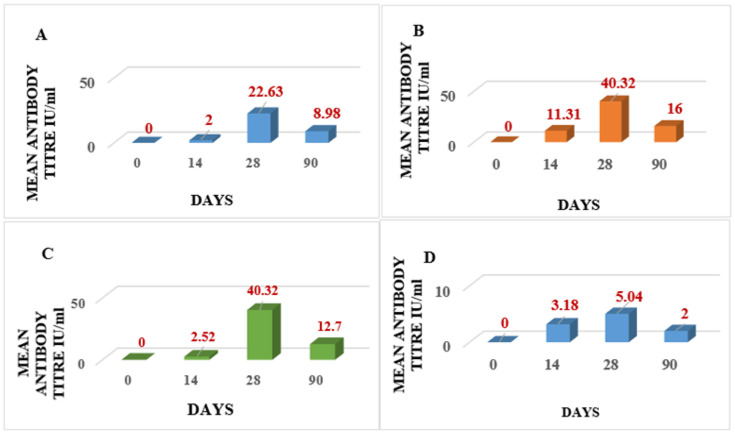
Geometric mean of RVNA titers (IU/mL) in cattle. (**A**) Group I vaccinated with rabies pre-exposure prophylactic vaccine administered through intramuscular route with booster dose. (**B**) Group II vaccinated with rabies pre-exposure prophylactic vaccine administered through intramuscular route without booster dose. (**C**) Group III vaccinated with rabies pre-exposure prophylactic vaccine administered through intradermal route with booster dose. (**D**) Group IV vaccinated with rabies pre-exposure prophylactic vaccine administered through intradermal route without booster dose.

**Figure 5 vaccines-11-00885-f005:**
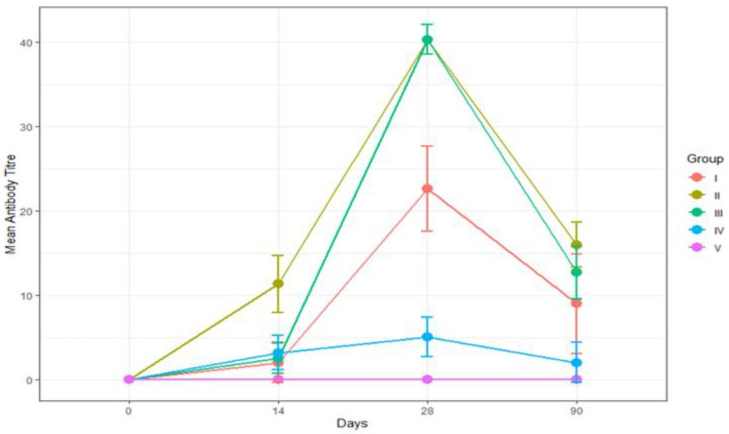
Comparison of geometric mean with geometric standard deviation of RVNA titers of Groups I, II, III, IV, and V.

**Figure 6 vaccines-11-00885-f006:**
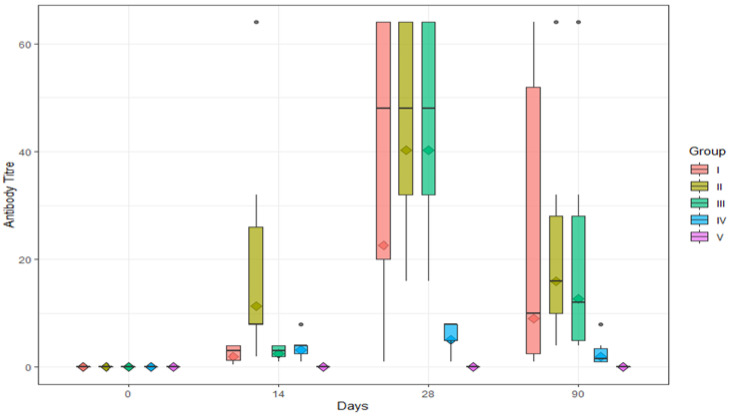
Box plot summarizing the range of RVNA titers in different vaccination groups tested at different time lapses (days). Notes: 1. Upper and lower boundaries of the box represent the 75th and 25th percentiles of the data range. 2. The thick horizontal line in the box represents the median. 3. The ♦ represents the average (geometric mean). 4. Longitudinal lines or whiskers represent the lower and upper limits of normal data distribution. 5. The ⦁ represents outlying data points.

**Table 1 vaccines-11-00885-t001:** Comparison of RVNA titers on different days in cattle of Groups I, II, III, and IV vaccinated with rabies pre-exposure prophylactic vaccine administered through different routes (with Group V as control).

Group and Route of Administration	Dose and Route	Day 0	Day 14	Day 28	Day 90
Group I (IM with booster dose)	1 mL IM	0 *	0.5–4	1- 64	1–64
Group II (IM without booster dose)	1 mL IM	0 *	2–64	16–64	4–64
Group III (ID with booster dose)	0.2 mL ID	0 *	1–4	16–64	4–64
Group IV (ID without booster dose)	0.2 mL ID	0 *	1–8	1–8	1–8
Group V (Control)	-	0 *	-	-	-

* A value of 0 was considered for titers < 0.5 IU/mL.

**Table 2 vaccines-11-00885-t002:** Geometric mean (GM) ± geometric standard deviation (GSD) of RVNA titers in cattle of Groups I, II, III, IV, and V.

Group	Dose & Route	RVNA Titer (IU/mL)			
		Day 0	Day 14	Day 28	Day 90
Group I (IM with booster dose)	1 mL IM	0 ± 0 ^A^	2.00 ± 2.40 ^A^	22.63 ± 5.08 ^Ba^	8.98 ± 5.91
Group II (IM without booster dose)	Iml IM	0 ± 0 ^A^	11.31 ± 3.39	40.32 ± 1.76 ^Ba^	16.00 ± 2.67
Group III (ID with booster dose)	0.2 mL ID	0 ± 0 ^A^	2.52 ± 1.76 ^A^	40.32 ± 1.76 ^Ba^	12.70 ± 3.1
Group IV (ID without booster dose)	0.2 mL ID	0 ± 0	3.18 ± 2.05	5.04 ± 2.32 ^b^	2.00 ± 2.40
Group V (Control)	-	0 ± 0	-	-	-

Note: Geometric mean ± GSD bearing different superscript letters within a column (upper case) and within a row (lower case) are significantly different at *p* < 0.05.

## Data Availability

The data presented in this study is available with in the article. There is no additional data to share.

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
