# Peer review of "Comparative Evaluation of Intradermal vis-à-vis Intramuscular Pre-Exposure Prophylactic Vaccination against Rabies in Cattle"

_vaccines, 2023, doi:10.3390/vaccines11050885_

Round 1

Reviewer 1 Report

This MS has to rewrite as Brief Note. Introduction, to describe how is cattle situation in India, how many animals are estimated, which is the main transmitter for rabies. Is really necessary to vaccinate cattle?. 

Methods, how is the titre of the whole dose vaccine and which is the titre in 0.2 mL?

Results. To explain Plate 1-3, in case Plate 2 what is that intradermical tissue?, in this case is difficult to believe that the vaccine produced this fluorescence level. 

Discussion, To reduce discussion in order to concretize the immune response  via ID and in cattle. 

Reviewer 2 Report

The article reports the comparison between IM and ID administration protocol of rabies vaccines in cattle in India.

As request in my comments, please better motivate the reason for this study and describe the differences with very similar works of the past (some of which are also Indian).

The paper can be published after a careful review of some parts of the paper, above all of references both in Introduction and in Discussion parts, some corrections, adding, clarifications and English language. 

For my complete comments see attached document 

Reviewer 3 Report

This is an interesting article investigating and comparing the administration of rabies vaccine in cattle via IM and ID routes and evaluating the immune response post vaccination. The titres were above adequate level (≥ 0.5 IU/mL) on day 14 and maintained up to 90 days in all animals administered rabies vaccine through IM and ID route with or without booster dose. The study indicated that both the routes of vaccine administration are safe and effective in providing protection against rabies. Both routes can then be considered for pre-exposure prophylaxis. However, ID route proved to be more economical due to its dose sparing effect.

Minor changes are indicated in the text.
